# Suppressing a Blocked Balance Recovery Step: A Novel Method to Assess an Inhibitory Postural Response

**DOI:** 10.3390/brainsci13101488

**Published:** 2023-10-21

**Authors:** David A. E. Bolton, Charlie C. Baggett, Chase A. Mitton, Sara A. Harper, James K. Richardson

**Affiliations:** 1Department of Kinesiology and Health Science, Utah State University, Logan, UT 84322, USA; a02223158@usu.edu (C.C.B.IV); chasemitton@gmail.com (C.A.M.); sara.harper@uah.edu (S.A.H.); 2Sorenson Center for Clinical Excellence, Utah State University, Logan, UT 84322, USA; 3Kinesiology Department, The University of Alabama in Huntsville, Huntsville, AL 35899, USA; 4Department of Physical Medicine and Rehabilitation, University of Michigan, Ann Arbor, MI 48109, USA; jkrich@med.umich.edu

**Keywords:** response inhibition, reactive balance, ReacStick, stepping

## Abstract

Stepping to recover balance is an important way we avoid falling. However, when faced with obstacles in the step path, we must adapt such reactions. Physical obstructions are typically detected through vision, which then cues step modification. The present study describes a novel method to assess visually prompted step inhibition in a reactive balance context. In our task, participants recovered balance by quickly stepping after being released from a supported forward lean. On rare trials, however, an obstacle blocked the stepping path. The timing of vision relative to postural perturbation was controlled using occlusion goggles to regulate task difficulty. Furthermore, we explored step suppression in our balance task related to inhibitory capacity measured at the hand using a clinically feasible handheld device (ReacStick). Our results showed that ReacStick and step outcomes were significantly correlated in terms of successful inhibition (r = 0.57) and overall reaction accuracy (r = 0.76). This study presents a novel method for assessing rapid inhibition in a dynamic postural context, a capacity that appears to be a necessary prerequisite to a subsequent adaptive strategy. Moreover, this capacity is significantly related to ReacStick performance, suggesting a potential clinical translation.

## 1. Introduction

Falls are a leading cause of fatal and nonfatal injuries among older adults, and many of these falls arise when people encounter complex settings with clutter or uneven terrain [1]. In such environments, factors such as processing speed and leg strength may be insufficient to avoid a fall. Here, the ability to override and suppress instinctual reactions—inhibitory control—has great relevance to controlling balance in the choice-demanding and often unpredictable situations we encounter daily. The value of inhibitory control in maintaining balance, though counterintuitive, is indirectly shown by the fact that performance on tests of response inhibition is related to falls, even in seemingly healthy older adults [2,3]. However, current knowledge is limited to the fact that inhibitory control and falls are correlated without any mechanistic understanding of how this particular capacity contributes to preserving postural equilibrium. While the cause of falls is multifactorial, understanding how inhibitory control impacts our ability to avoid falling is significant because the ability to inhibit automatic, but unwanted action could offer a powerful mechanism to prevent falls in the complex scenarios faced every day. Indeed, evidence suggests that inhibitory capacity independently predicts successful balance recovery beyond established fall risk factors such as strength, sensory acuity, and processing speed [4]. 

To avoid falling, muscles throughout the body must be quickly coordinated to resist postural perturbation [5]. Video footage from long-term care facilities has shown that stepping after imbalance is the predominant response, usually with multiple steps, highlighting the importance of step reactions [6]. However, many real-life scenarios require the adaptation of an automatic step, and short-latency inhibitory control is particularly relevant to the control of balance when a compensatory step must be suppressed to prevent further instability. To illustrate, consider the challenge that must be overcome when standing on a crowded bus. An unexpected braking of the bus would elicit rapid postural responses to regain stability. If the disruption to stance is great enough, the only option for recapturing the displaced center of mass involves a change of support, such as a forward step. However, what if this step is blocked by an obstacle (e.g., groceries at your feet)? This context might then demand abrupt cessation of the automated stepping reaction, supplanted instead by reaching the hand towards an available handrail.

Stepping to recover balance is an important strategy we use to avoid a fall [7,8], but when faced with obstacles in the step path, we must adapt such planned step reactions. Prior work suggests that active inhibition of a movement or action is a necessary and distinct process before the execution of an alternative, more adaptive movement or action [9]. In this sense, inhibition is a key process underlying behavioral flexibility. Regarding gait/balance, preplanned steps or movements must be inhibited with great efficiency following an unanticipated environmental change causing postural instability to allow for the generation of a new, adaptive strategy within the few hundred milliseconds available before a fall occurs [10,11]. Recently, our research team published a novel method emphasizing the ability to inhibit a balance recovery step by signaling participants to abort a step when presented with a tone cue [12]. Our method was modelled after the stop signal task, a classic measure of action cancellation where participants suppress a response when presented with a stop cue [13,14]. The current approach extends from our previous method by using a *visual* rather than auditory signal to prompt step suppression. Specifically, we imposed an obstacle to block a step path, thus offering a more behaviorally relevant visual cue to emphasize response inhibition in a reactive balance context. Therefore, the primary aim of this study was to develop and then describe our new reactive balance task.

In addition to describing our new method, we also explored the relationship between step suppression and performance on a test of rapid inhibition using a hand-based assessment tool called a ReacStick. The ReacStick is a clinically feasible handheld device that evaluates rapid inhibition and has been found to predict mobility-based outcomes [4,15]. For example, Okubo et al. [4] measured several standard fall risk variables such as leg strength, postural sway, simple and choice reaction time, etc., in relation to performance in a laboratory-based perturbation task. In that study, participants needed to adapt their gait to prevent a fall, and the strongest predictor of balance recovery was rapid response inhibition accuracy measured using the ReacStick. In another study by the same researchers, ReacStick performance was associated with frontal plane gait variability when people crossed a chaotic and uneven surface [16]. These participants represented a spectrum of neuromuscular function, and once again, other lower-limb neuromuscular attributes were uninformative in predicting stability outcomes beyond response inhibition accuracy [16]. Therefore, by including this exploration between step suppression in a balance recovery task and ReacStick outcomes, we aimed to provide clinical translation of our laboratory findings.

## 2. Materials and Methods

### 2.1. Participants

A convenience sample of 21 young adults aged 18–30 (12 female) were recruited; they provided informed written consent prior to participation in this study. Participants were excluded if they had a neurological illness, traumatic brain injury such as concussion within the past 6 months, were unable to stand for the duration of the reactive balance test (2 h), or a recent musculoskeletal injury that interfered with testing. Data from two participants were excluded due to equipment malfunction. Procedures were approved by the Utah State University Institutional Review Board and conducted in accordance with the Declaration of Helsinki.

### 2.2. Data Acquisition

#### Force Plates

Three force plates (Kistler Instrument Corp., Winterthur, Switzerland) measured vertical ground reaction forces to detect a step response (e.g., lift-off and touchdown). Two smaller plates measured forces in the start position with one force plate under each foot. The third plate was in front of the participant to capture touchdown of a step. Signal software v.7. (Cambridge Electronic Design, Cambridge, UK) sampled data at 1000 Hz.

### 2.3. Test Procedures

#### 2.3.1. Lean and Release Reactive Balance Test with Leg Block

Forward perturbations were imposed using a custom-made lean-and-release system [12,17] while liquid crystal occlusion goggles (Translucent Technologies Inc., Toronto, ON, Canada) controlled visual access, as shown in Figure 1. Participants were supported in a forward lean of about 6° anterior rotation at the ankle using a body harness attached to a support cable secured to the wall using a magnet. Participants fixated their gaze approximately 1.2 m ahead but adjusted as needed to ensure the leg block was visible when placed in the stepping path. The magnet produced a time-specific release from this supported lean, causing a forward perturbation. This cable release acted as a STEP cue in most (80%) trials, where participants were instructed to step forward as quickly as possible. In 20% of trials, a leg block was positioned to obstruct the step path. For these STOP trials, participants were told to suppress a step and relax into a secure secondary catch cable. This catch cable was slightly longer than the support cable, allowing a forward fall of approximately 10° before arrest. The 80:20 ratio biased the STEP response, forcing participants to suppress a prepotent step when the step path was blocked. Occlusion goggles opened near the start of each trial to reveal the specific response condition (STEP or STOP) with a range of visual preview delays (VPD), i.e., the delay between goggles opening and cable release. An analog–digital recorder and Signal software (Power 1401-3A, Cambridge Electronic Design, Cambridge, UK) were used to control cable release, open/close occlusion goggles, and drive servo motors to move the leg block. A failsafe support cable was attached to the ceiling to ensure participant safety and leg blocks were made with a compliant material secured onto the servo motors with movable hinges (servo savers) to prevent impact injury.

#### 2.3.2. ReacStick

The ReacStick is a handheld device that measures short latency response inhibition and reaction time, as shown in Figure 2, with details of the task described previously [15]. Briefly, this test uses a ‘ruler-drop’ paradigm to measure simple reaction time, and go/no-go reaction accuracy under time pressure (i.e., the decision to grasp the stick or let it fall must occur in less than 400 ms) [15]. For reaction accuracy, participants must decide to catch or let the stick fall based on a random illumination of lights affixed to the device. The challenge here is to overcome a natural urge to grasp the falling stick when lights do not illuminate. The outcome for reaction inhibition is the percentage of 10 light off trials appropriately not caught, providing an index of response inhibition accuracy under temporal constraint. The outcome for reaction accuracy is the percentage of the sum of appropriately caught light on trials and appropriately not caught light off trials divided by the total number of 20 trials. An advantage with this test is an emphasis on the time pressure to respond, similar to the absolute time pressure to prevent a fall. This is distinct from other cognitive tests, where people commonly slow down to avoid mistakes.

#### 2.3.3. Experimental Protocol

Each test session started with ReacStick testing, which lasted approximately 10 min. Next, participants were positioned in the lean and release system where they were familiarized with the balance task. To begin each trial, participants leaned into a support cable keeping both feet in contact with the floor. Each trial started with a randomized delay before cable release. Participants practiced by first performing a rapid step following cable release (Go cue). They were told that this rapid step was the default response and were allowed to step with either leg but asked to consistently use the same leg throughout testing. After cable release, participants returned to the designated start position on the platform, outlined with marker on force plates, and the magnet reattached. They were instructed to remain relaxed and react only when the cable was released. Next, participants practiced the STOP condition where the leg block was positioned in front of both legs. As before, participants were released from the support cable, but this time, they were instructed to relax and let themselves fall forward into the secondary catch cable. Following exposure to both the STEP and STOP conditions, participants then practiced the actual task, where STOP and STEP cues were randomly intermixed, performing 5–10 practice trials. At this stage, formal testing began. Participants were reminded to step as quickly as possible upon cable release but to relax and fall forward when the leg block appeared. Each trial lasted five seconds, followed by a five- to ten-second break to reset the start position. Testing involved 200 total trials and lasted approximately two hours. 

While developing our method, our goal was to determine a challenge level that was difficult, yet manageable for most participants to avoid any ceiling or basement effects. To accomplish this, we increased the level of task challenge in a stepwise fashion by shortening the VPD in successive test groups. We manipulated the VPD to control how much time was available for participants see if a block was present or not with the underlying assumption that a shorter VPD represents a more difficult challenge. The first two participants experienced an easier version of this task with VPDs ranging 50–200 ms with randomized delays of 50 ms, 100 ms, 150 ms, and 200 ms and an equal number of trials at each VPD (i.e., 40 total STOP trials = 10 trials per VPD with 4 levels of VPD). Given the nearly flawless performance of these first two participants, the level of difficulty was increased for the next few participants using randomized delays of 25 ms, 50 ms, 75 ms, and 100 ms. To increase task difficulty even further, the next group was presented with a range of delays, −25 ms, 0 ms, 25 ms, and 50 ms, which included very challenging situations where the cable release was simultaneous with vision (0 ms) or even slightly before (−25 ms). Most individuals demonstrated at least some stepping errors at this stage; therefore, we continued testing all remaining participants at this level of difficulty. This is consistent with past research into the use of vision during a balance recovery step finding that when vision was limited to the onset of perturbation, clear deficits emerged in the step reaction compared to situations where visual spatial information could be accrued in advance [18].

### 2.4. Data Analysis

To determine whether a step was taken, force plates quantified the vertical ground reaction force beneath the stepping leg in the lean and release task. Force plate data were smoothed (Signal software, Cambridge Electronic Design, Cambridge, UK) and then exported as a text file where they were analyzed using a customized LabVIEW program (National Instruments, Austin, TX, USA). Performance outcomes of the balance test were described in terms of average and standard deviations. For the exploratory analysis into the relationship between the balance and ReacStick outcomes, bivariate correlations were used. 

#### 2.4.1. Force Plate Analysis

Liftoff from the force plate beneath the stepping leg was classified as a response error on STOP trials, with liftoff defined as the moment vertical force reached zero. Premature step trials (defined as weight shifts >10% of baseline within 50 ms of cable release) were eliminated. The outcome measures of interest on the balance task were (a) inhibition accuracy (IA) and (b) reaction accuracy (RA). The first measure, IA, was defined as successful step suppression during STOP trials expressed as a percentage of the total number of STOP trials. The second measure, RA, was defined as a combination of correctly suppressed steps during STOP trials plus correct STEP trials (i.e., no Go omissions) expressed as a percentage of the total number of trials. The key distinction between these outcome measures is that IA represents inhibitory control directly, whereas RA is intended to reflect global performance in a task that occasionally demands response inhibition.

#### 2.4.2. ReacStick Analysis

The outcome measures of interest on the ReacStick task were similar to the balance task with IA and RA. In this case, IA represents successful grasp suppression on ReacStick lights off trials (i.e., trials where participants are cued to suppress a grasp and let the device fall). RA expresses the percentage of correct responses on both ‘Go’ and ‘No Go’ trials by combining correct steps/grasps with correctly inhibited steps/grasps, which is then divided by the total number of trials to provide an overall measure of task success.

#### 2.4.3. Exploratory Analysis

While the main aim of this study was to develop and describe our novel method for assessing response inhibition in a reactive balance context, a secondary aim was to determine if performance during the balance task was associated with ReacStick performance. Only the 12 participants with balance data using the most challenging version of VPD were included in this exploratory analysis. The first two VPD groups were deemed insufficiently challenging and prone to ceiling effects given high success rates across all five participants (see Table 1). For these 12 participants, Pearson correlations tested the relationship between the balance task performance and ReacStick performance, separately for IA and RA (*p* < 0.05). 

## 3. Results

The average reaction time of STEP trials was 320 ± 23 ms across all VPD conditions with 1.2% omission errors on average (0–5% range). Key outcome variables from the balance task and ReacStick are provided in Table 1. As described in the Section 2, the degree of challenge was progressively increased across three different VPD levels until we arrived at a level that resulted in an average stopping success rate of 73% (33–95% range). A one-tailed, bivariate correlation tested the association between successful ReacStick performance and success of the balance task. There was significant correlation for the global measure of task success (i.e., reaction accuracy) r = 0.76; *p* = 0.002, and successful inhibition (i.e., inhibition accuracy) r = 0.57; *p* = 0.026 (Figure 3).

## 4. Discussion

The purpose of this study was to develop a reactive balance test emphasizing response inhibition using a visible obstruction as a stop cue. We adjusted the amount of time participants had to view the response environment before deciding to step or not, and using an iterative process, we arrived at a challenge level most participants found difficult, yet manageable. This new method may offer a complementary way to investigate postural control by stressing a key cognitive capacity (response inhibition) related to fall risk. Notably, this capacity is largely unaccounted for in many of the standard postural assessments such as the Berg balance scale [19], the BESTest [20], Physiological Profile Assessment [21], and dynamic posturography [22].

Inhibition is foundational to behavioral flexibility, and when one motor plan becomes obsolete, a new motor plan is needed. Critically, at the instance of perturbation, the current motor plan must be inhibited before a new plan can be executed. Overall, our new method represents an important way to reveal a key mechanism that applies in any situation where behavioral flexibility is necessary to avoid a fall. The proposed method for testing inhibitory control in a reactive balance context uses stringent experimental controls to isolate inhibitory capacity and in research settings this could be used to identify specific neural mechanisms that predict successful balance reactions (e.g., with the aid of neuroimaging tools such as functional near-infrared spectroscopy or electroencephalography). There is clear evidence showing that an unexpected event, such as a postural perturbation, quickly recruits a broad neural stopping network [9], which is required before executing a new motor plan to allow for balance recovery. In daily life, the abrupt cessation of conversation by a person who suddenly slips illustrates this concept. Our expectation is that this rapid shutdown mechanism provides the foundation for postural adaptations in the myriad settings we face daily, and this inhibitory capacity ultimately applies in any situation where behavioral flexibility is necessary to avoid a fall.

In the present paper, we extend from our earlier published method, where an auditory stimulus compelled suppression of an automatic balance recovery step [12]. However, now we use a leg block to prompt step suppression in a more naturalistic and functionally relevant way. Our first version, using a stop tone, was based on a traditional cognitive neuroscience task known as the stop signal task [23], where an auditory cue instructs participants to suppress an automatic response (usually a button press). When we first developed our method to emphasize response inhibition in a reactive balance task, an auditory signal offered an easy way to deliver a STOP cue relative to a Go cue (i.e., cable release). By contrast, our current method involves maneuvering a leg block into position coordinated with opening/closing of occlusion goggles, all of which requires extra equipment and programming considerations. However, this new method also has the advantage of greater ecological validity by using a visual STOP cue (i.e., an obstacle in the step path); a cue that is directly relevant to balance recovery. Overall, this means that each method comes with different advantages that can be weighed out depending on specific needs.

For participants that completed the most difficult VPD level, we also evaluated the association between step inhibition and ReacStick performance, revealing a significant positive correlation. Our findings are consistent with past research that demonstrated ReacStick performance was the strongest predictor of balance recovery after slip/trip and gait variability on an uneven walking surface, even when accounting for factors such as leg strength and simple processing speed [4,16]. In older cancer survivors, increased fall risk is often attributed to peripheral neuropathy without considering the cognitive impact of chemotherapy, and yet recent research shows that poor inhibitory control measured with the ReacStick is associated with impaired balance and predicts future falls independent of peripheral nerve status [24]. Therefore, inhibition measured with this handheld device appears to generalize and evaluates a cognitive capacity, namely, short latency motor inhibition, which is critical for maintaining postural stability [25]. Also, the correlation in ReacStick reaction accuracy indicates that the ability to generate accurate responses overall (i.e., stopping or going when cued to do so) generalizes across these tasks where occasional inhibition is required. Interestingly, the step inhibition and ReacStick tasks were both challenging in this young, healthy cohort, suggesting that these methods are sufficiently sensitive to discriminate between older people at increased fall risk due to delayed motor inhibition and those who are not. These results are also consistent with past work, revealing a correlation between performance on a reactive balance task requiring step suppression and performance on a seated stop signal task where finger responses were used (i.e., keystrokes) [26]. Given emerging evidence linking executive processes such as inhibition with fall prevalence [11,12], assessment techniques evaluating rapid inhibitory motor control may address an important gap in fall risk prevention.

A recent review evaluated response inhibition as part of controlling balance in older populations and observed that despite heterogeneity in tasks and outcome measures, all of them revealed the value of inhibition in postural performance [27]. Specifically, the authors focused on studies where inhibitory control was directly required to succeed in the postural task versus dual-tasking (i.e., concurrent, but separate cognitive and postural tasks), which is a critical distinction. Consistent with this idea, our new method demands inhibition of a highly prepotent balance recovery step to successfully accomplish the task, suggesting a way to incorporate inhibition directly into balance assessment, at least in a research setting. A reactive balance test that emphasizes inhibition would be of particular value in populations where inhibitory control is compromised, and this includes otherwise healthy older adults where age-related deficits have been noted in tasks that require action cancellation [28]. With a diverse range of factors that contribute to falls, there is a critical need for the identification of the specific impairment(s) leading to falls, and performance on a test of response inhibition has diagnostic potential as a novel, early marker of fall risk. Targeted treatment and risk mitigation for older people with diminished inhibitory control would focus on optimizing cognitive health through reduction in polypharmacy and addressing sleep/anxiety/depression and metabolic disorders known to impact inhibitory function. This stands in stark contrast to the usual referral of patients who fall to nonspecific physiotherapy. Additionally, several research teams have shown promising results using perturbation-based training as a means of improving fall resistance in older adults. The inclusion of training scenarios that call for response inhibition and behavioral adaptation into a context of postural threat could lead to avoiding a fall in a real-world scenario. Recent highly powered studies point out significant limitations to current practice for reducing falls [29,30] and identify a need for better concepts to increase efficacy of interventions [31]. Given age-related deficits in cognitive function, this element of control could have a pronounced effect on the cause of falls, best exposed in settings that demand response inhibition.

As a limitation, we acknowledge the artificial nature of our balance assessment, where participants were perturbed from a leaning start position and where vision was manipulated using occlusion goggles. Additionally, participants were aware of certain task elements in advance, including the direction and magnitude of perturbation, which is unlike how we experience falls in the real world. Here, we contend that our modified lean-and-release procedure, though an unnatural experience, is appropriate for precisely manipulating events and isolating a need to inhibit action during the experience of falling. Indeed, the all-or-none nature of our task—step or relax—is intended to replicate traditional cognitive assessments such as the stop signal task, where participants either press a button or withhold a response altogether [23]. Our goal was to carefully control threats to internal validity in order to emphasize response inhibition of a balance recovery step with control over timing of a visual stop cue. Such rigid control in lab settings often contrasts real-world generalization; however, this constrained approach is necessary at an early stage of investigation to identify inhibitory control mechanisms in reactive balance. Finally, we recognize that our participant number, particularly with reference to the postural response/ReacStick correlations, was relatively low and a larger sample size would be required to make any definitive claims on this relationship. Accordingly, despite their robust nature, our results should be viewed as preliminary.

## 5. Conclusions

In the present paper, we described a new laboratory behavioral assessment that stresses inhibitory capacity. By using a physical step obstruction and manipulating access to vision, we offer a novel way to emphasize inhibition in a reactive balance task. A particularly exciting aspect of our findings is the relationship between performance on the balance task with performance on a simpler, handheld assessment, i.e., the ReacStick test. Given that ReacStick rapid inhibition accuracy has been shown to predict fall prevalence in a laboratory-based setting where participants were exposed to slips and trips [4], such generalization may offer immediate clinical translation given its ease of use [25].

## Figures and Tables

**Figure 1 brainsci-13-01488-f001:**
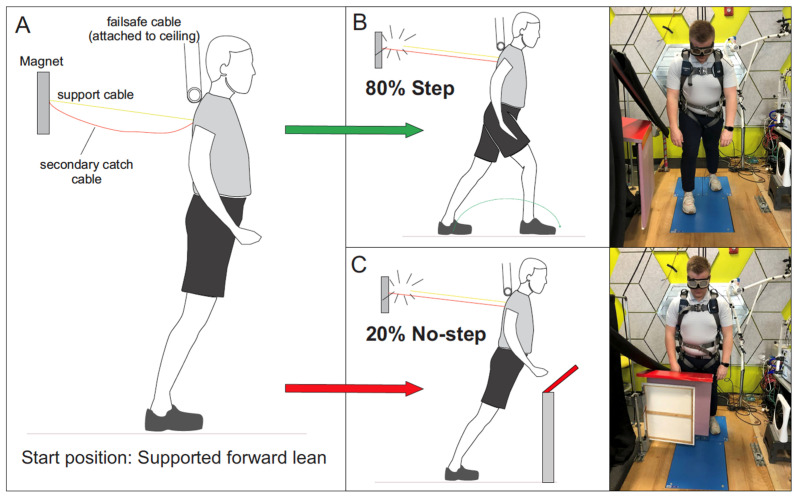
Lean and release task. (**A**) Supported lean before support cable release (yellow solid line). A failsafe cable secured the participant to the ceiling to prevent participants from falling to the ground. (**B**) Forward step following cable release in 80% of trials, referred to as the STEP condition. (**C**) Leg block cueing participants to prevent a step and relax in 20% of trials, referred to as the STOP condition. For STOP trials, a secondary catch cable (orange line) prevented a fall. The onset of vision relative to cable release was controlled with liquid crystal occlusion goggles and this relative timing was referred to as the visual preview delay, or VPD. Three sets of VPD were used in different groups as follows: (1) 50 ms, 100 ms, 150 ms, and 200 ms; (2) 25 ms, 50 ms, 75 ms, and 100 ms; (3) −25 ms, 0 ms, 25 ms, and 50 ms. Group three posed the greatest challenge, with very little time allowed between the goggles opening and cable release. This group even included a negative delay, i.e., cable released 25 ms *before* vision. Adapted with permission [12].

**Figure 2 brainsci-13-01488-f002:**
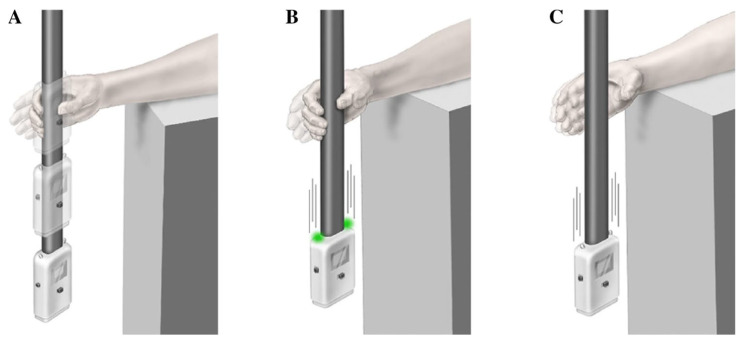
ReacStick task. The ReacStick is a handheld device that measures short latency response inhibition and reaction time, as shown in the bottom three panels. (**A**) Simple reaction time test where the device is released, and the participant needs to catch it as quickly as possible (not included). (**B**) Reaction accuracy test showing the condition where lights randomly illuminate as the device is released, which is the indicator for the participant to catch the device. (**C**) Reaction accuracy test showing the condition where lights do not illuminate upon release and the participant must resist the urge to catch it. Note that the decision to catch or not must be made before the device strikes the ground when dropped from desk height (approximately 360 ms). The bottom image was adapted with permission from [13].

**Figure 3 brainsci-13-01488-f003:**
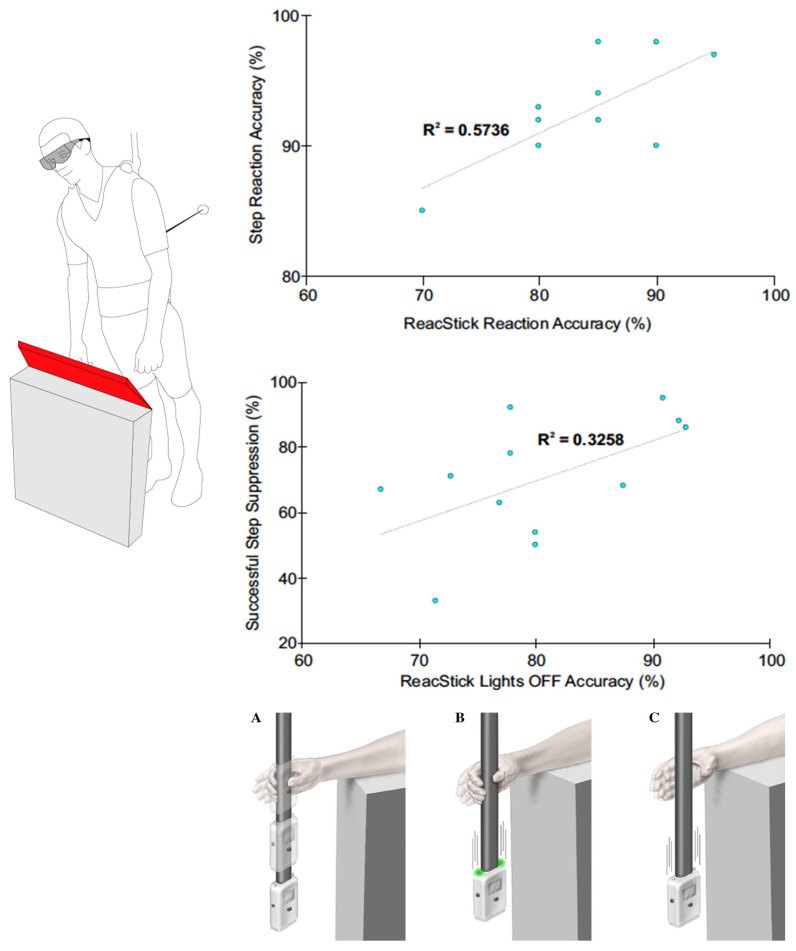
Scatterplot depicting significant positive correlations between balance (*y*-axis) and ReacStick (*x*-axis) outcomes. Top graph: Reaction Accuracy—Expresses the percentage of correct responses of both ‘Go’ and ‘No Go’ trials by combining correct steps/grasps with correctly inhibited steps/grasps, which is then divided by the total number of trials to provide an overall measure of task success. Bottom graph: Inhibition Accuracy—Successful balance recovery step suppression of STOP trials is shown in relation to successful grasp suppression of ReacStick lights off trials (i.e., trials where participants are cued to suppress a grasp and let the device fall). The bottom graph is strictly focused on successful inhibition, where participants are cued to suppress either a step or grasp. ReacStick task conditions: (**A**) Simple reaction time test, (**B**) Reaction accuracy test showing the condition where lights randomly illuminate to cue a grasp, and (**C**) Reaction accuracy test showing the condition where lights do not illuminate and the participant must resist the urge to catch it.

**Table 1 brainsci-13-01488-t001:** Outputs are organized by visual preview delay (VPD) into three groups: (1) 50/100/150/200 ms, (2) 25/50/75/100 ms, (3) −25/0/25/50 ms with averages and standard deviation (SD). The average reaction times (RT) are measured in milliseconds (ms) for frequent ‘Go’ responses (step or grasp) along with the percentage of successful stops in both tasks. Overall task performance is presented as reaction accuracy (RA), while successful stopping is presented as inhibition accuracy (IA). For ReacStick outcomes, the latency to accurately grasp the device represents the average grasp reaction time when the lights came on. Note: Two participants listed here had stepping data but no ReacStick data.

		Balance			ReacStick		
VPD	Subject	IA (%)	RA (%)	RT (ms)	IA (%)	RA (%)	RT (ms)
1	1	89	97	385	92	85	228
	2	100	99	321	90	90	231
	Average	**95**	**98**	**353**	**91**	**88**	**230**
	SD	8	1	45	1	4	13
2	3	98	95	310	100	100	280
	4	81	96	284	86	90	212
	5	90	98	331	83	80	224
	Average	**90**	**96**	**308**	**90**	**90**	**239**
	SD	9	2	24	9	10	5
3	6	68	92	339	88	85	212
	7	88	98	327	92	90	210
	8	67	92	313	67	80	223
	9	93	97	356	-	-	-
	10	80	96	279	-	-	-
	11	63	93	328	77	80	199
	12	78	94	322	78	85	202
	13	92	98	310	78	85	219
	14	50	90	320	80	80	222
	15	33	85	283	71	70	211
	16	95	97	333	91	95	245
	17	86	97	317	93	95	205
	18	54	90	308	80	90	213
	19	71	92	307	73	85	214
	Average	**73**	**94**	**317**	**81**	**85**	**221**
	SD	18	4	20	9	7	10

## Data Availability

The data presented in this study are openly available in the Open Science Framework at osf.io/crdw3 (accessed on 14 October 2023).

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
