# Peer review of "Suppressing a Blocked Balance Recovery Step: A Novel Method to Assess an Inhibitory Postural Response"

_brainsci, 2023, doi:10.3390/brainsci13101488_

Round 1

Reviewer 1 Report

The paper presents an interesting method to assess an inhibitory postural balance.  For their research, the authors used a handheld device ReacStick, Force plates, liquid crystal occlusion goggles and a custom-made lean-and-release system. All test procedures, equipment and analysis of obtained data are described in detail and clearly.

It is very gratifying that this is a continuing work of scientists and it is very welcome. When reading the work, one gets the feeling that they know perfectly what they are doing and what they expect, perhaps because of this, they assess the limitations of the results presented no less accurately and openly. I don't want to comment or criticize something, I only invite authors to continue the work.

Here are some minor comments:

-        Conclusions in this case are more debatable and should be specific facts from the results obtained only. I recommend specifying the conclusions and leaving the discussion part for discussion.

-        Another note would be for the presentation of data analysis: how many samples were obtained in total, what was the data distribution, and why was one or another statistical analysis method chosen? You can guess even after seeing the results, but if they were written, it would be much clearer for the reader and it would be easier to trust the presented results.

I really think the publication has potential, even if the results should be considered preliminary. It was a pleasure to read and I wish the authors success in their research.

Reviewer 2 Report

The paper appears to have a solid foundation and presents a novel approach to assessing postural balance with an emphasis on response inhibition. The methodology, while innovative, has been clearly explained, and the results align with the study's objectives. The authors have also conscientiously acknowledged the limitations, indicating a thorough analysis.

However, a few areas might benefit from refinement:

1. Expand on the external validity: Addressing the concerns about the artificial nature of the balance assessment in more depth would be beneficial. It would help if there were comparisons or references to how this artificial setting might translate to real-world scenarios.

2. Sample size: Even though the authors acknowledge the limited sample size, a brief discussion or future intention of conducting a larger-scale study would provide clarity.

3. Clarifications on methodology: A more in-depth discussion about the choice of visual stimuli, as well as the decision to transition from auditory to visual cues, might enhance the reader's understanding.

4. Expansion on Practical Implications: The paper does touch upon the potential clinical applications, especially with the mention of the ReacStick test. However, delving deeper into how these findings can be applied in real-world clinical settings, rehabilitation centers, or even in designing public spaces for the elderly might provide more value for a wider range of readers. Outlining the broader societal or health implications could make the paper more impactful.

Reviewer 3 Report

Overall, the manuscript appears to be sufficiently well-structured but a bit superficial and presents some conceptual and formal shortcomings.
The introduction should better explain the ratio of the experiment and explain some concepts of inhibitory motor control.
In the Introduction, line 42 is a bit in contrast with line 32.
In Materials and Methods, I suggest quantifying the familiarization task by including the number of repetitions. Furthermore, the only statistical analyses concern ReacStick, so I suggest writing or implementing others regarding the Lean and release balance test.
In Results, lines 227 to 230 are a repetition of the experiment protocol, so I suggest moving the percentages of success rate there and deleting this sentence.
There is also a repetition in lines 231–234. I suggest deleting this sentence.
In Figure 2, the bottom image should be moved to the materials and methods paragraph, maybe in Figure 1.
Finally, both the discussion and the conclusion are a bit superficial, and they should be improved.

Round 2

Reviewer 3 Report

I appreciate the changes made to the manuscript. The text is much more clear and correct.